# Health of children who experienced Australian immigration detention

**Shidan Tosif**[1,2,3]*, **Hamish Graham**[1,2,3], **Karen Kiang**[1], **Ingrid Laemmle-Ruff**[2], **Rachel Heenan**[1], **Andrea Smith**[1], **Thomas Volkman**[1,3], **Tom Connell**[1,3], **Georgia Paxton**[1,2,3]

1 Department of General Medicine, Royal Children's Hospital Melbourne, Parkville, Victoria, Australia,
2 Infection and Immunity, Murdoch Children's Research Institute, Melbourne, Victoria, Australia,
3 Department of Paediatrics, The University of Melbourne, Melbourne, Victoria, Australia

* shidan.tosif@rch.org.au

## Abstract

### Background

Australian immigration policy resulted in large numbers of children being held in locked detention. We examined the physical and mental health of children and families who experienced immigration detention.

### Methods

Retrospective audit of medical records of children exposed to immigration detention attending the Royal Children's Hospital Immigrant Health Service, Melbourne, Australia, from January 2012 –December 2021. We extracted data on demographics, detention duration and location, symptoms, physical and mental health diagnoses and care provided.

### Results

277 children had directly (n = 239) or indirectly via parents (n = 38) experienced locked detention, including 79 children in families detained on Nauru or Manus Island. Of 239 detained children, 31 were infants born in locked detention. Median duration of locked detention was 12 months (IQR 5–19 months). Children were detained on Nauru/Manus Island (n = 47/239) for a median of 51 (IQR 29–60) months compared to 7 (IQR 4–16) months for those held in Australia/Australian territories (n = 192/239). Overall, 60% (167/277) of children had a nutritional deficiency, and 75% (207/277) had a concern relating to development, including 10% (27/277) with autism spectrum disorder and 9% (26/277) with intellectual disability. 62% (171/277) children had mental health concerns, including anxiety, depression and behavioural disturbances and 54% (150/277) had parents with mental illness. Children and parents detained on Nauru had a significantly higher prevalence of all mental health concerns compared with those held in Australian detention centres.

**Data Availability Statement:** Data cannot be shared publicly because of sensitivity of information. De-identified data will be made available upon request to the Royal Children's

Hospital Human Research Ethics Committee, contact Alexandra Robertson at rch.ethics@rch.org.au.

**Funding:** The authors received no specific funding for this work.

**Competing interests:** A/Prof Paxton has provided advice to Department of Home Affairs (previously the Department of Immigration and Border Protection) through the Minister's Council for Asylum Seekers and Detention (2015-2018), the Home Affairs Independent Medical Advisors Panel (previously the Independent Health Advisors Panel, from 2014-ongoing) and the Health Subcommittee of the Joint Advisory Committee for Nauru Regional Processing (2013–2016). GP also chairs a working group on immunisation in refugee and asylum seeker populations for the Victorian Department of Health (from 2015). The Royal Children's Hospital Immigrant Health Service is funded by the Victorian Department of Health.

## Conclusion

This study provides clinical evidence of adverse impacts of held detention on children's physical and mental health and wellbeing. Policymakers must recognise the consequences of detention, and avoid detaining children and families.

## Introduction

Australia has had a policy of mandatory immigration detention for people arriving without a valid visa since 1992, and indefinite detention is possible under Australian law [1]. During 2009–2013, increasing numbers of asylum seekers travelled to Australia by boat, with more than 51,000 arrivals in this period [2]. Whereas earlier asylum seeker boat arrivals had predominantly been adult males travelling alone, these cohorts included large numbers of families with children. Australian government and policy changes from mid-2013 resulted in prolonged detention for asylum seeker boat arrivals, with large scale use of Immigration Detention Centres (IDC) on the Australian mainland and territories, and the use of offshore processing in Australian-contracted Regional Processing Centres (RPC) in the Republic of Nauru and on Manus Island, Papua New Guinea (PNG). Families were randomly detained in IDC or RPC, if they arrived in Australia by boat after August 2012.

At the end of 2014, legislative changes [3] led to large-scale releases from IDC on the Australian mainland and territories. A second cohort of children and families, who had been detained in the Nauru RPC and returned to Australian IDC for medical reasons over 2013–15, were released by the end of April 2016, after spending nearly 3 years in detention. A third cohort of children and families were medically transferred to Australia throughout 2018, after spending 5 years in Nauru. The last refugee and asylum children on Nauru were returned to Australia in November 2018.

After release from IDC, children and families transitioned to either 'community detention' (CD) or onto a bridging visa (BV) in the community. People in CD do not have a visa, work rights or access to Medicare (Australia's universal health insurance scheme), and they are subject to curfews. Bridging visas include access to Medicare, and included work rights (from 2015). However, BV were generally of short duration (2–6 months) during 2014–2017, and expired frequently, usually precluding employment. Both CD and BV are associated with considerable ongoing uncertainty. As of 2022, some asylum seeker children and families are still awaiting a decision on the outcome of their primary refugee protection claim [4, 5], under the (previous) Australian Government's 'fast-track' process.

Concerns have been raised about the impact of Australian immigration policy and detention on the health and wellbeing of children, both locally and internationally [6–11]. Australian IDC and RPC are secure facilities, with perimeter fencing, restrictions on movement and a visible security presence. Numerous authors and independent monitoring agencies have raised concerns about inadequate facilities for children and families, lack of privacy, frequent room checks and body searches, safety issues and child protection issues [12–14]. Delays in processing protection claims, lack of notification about transfers, the threat of offshore processing and deportation, and the unclear duration of detention contributed to profound uncertainty and consequent impact on mental health [15, 16]. Children in IDC/RPC are specifically affected by restrictions on their movement, play and education, exposure to violence and adults with severe mental illness, the impacts of poor parental mental health and breakdown of family structure, and the duration and timing of detention during critical periods of

their development [8, 17]. The risks of detention are amplified for children and families in off-shore RPC facilities [18]. The geographical location presents complexity for clinical care due to limited access to specialist medical services, and poor living conditions are compounded by environmental and infrastructure challenges [19].

There is limited evidence of information on the health status of children who have experienced held detention in Australia or globally. Differences in clinical outcomes arising from onshore or offshore held detention in Australia have not been described. Our aim is to describe the physical and mental health of children who experienced held detention and identify common themes and issues arising in clinical care.

## Methods

### Study design and participants

We conducted a retrospective cross-sectional study of asylum seeker children and adolescents who had experienced held detention, or were born to parents who had experienced held detention. This study was approved as a retrospective audit by the Royal Children's Hospital (RCH) Human Research Ethics Committee (HREC; Reference Number: 36386A). We defined held detention as any time detained in an IDC on the Australian mainland or territories (including Christmas Island) or in Nauru or Manus RPC, or time residing in Nauru due to offshore processing arrangements.

The RCH is a tertiary paediatric hospital in Melbourne, Australia. The RCH Immigrant Health Service is a central hub for provision of care to asylum seeker and refugee-background children and young people, with referrals from community providers, detention health providers (International Health and Medical Services—IHMS) and other RCH services. The Immigrant Health Service provides assessment and treatment for medical and mental health problems, and coordinates care with community based services (general practitioners, refugee and maternal child health nurses). The Immigrant Health Service includes paediatricians, mental health clinicians (psychiatrist, psychologist, nurse), social workers, dental therapist and clinic nurse coordinator. Children are referred to the Immigrant Health Service from a range of primary care services, and referrals are not restricted by time–children can be seen at any point from days to years after arrival. Children in detention at the time of their initial review were referred by IDC health services at the discretion of their primary health provider. Approximately 80% of attendances require interpreting assistance.

Eligible children referred to the RCH Immigrant Health Service from January 2012 to December 2021 were included if they were aged less than 18 years at the time of their first visit. Children who attended at least one appointment at the immigrant health clinic were included, even if they did not attend subsequent appointments or complete clinical assessment. Patients were identified through a search of the immigrant health electronic medical records (EMR—CAReHR™) and designated patient lists in the hospital-wide EMR implemented from 2016. We also reviewed clinic attendance records, searched hospital records using the address of the Melbourne immigration detention centre, and searched outpatient billing records for relevant billing codes.

### Variables

We extracted data from clinical records (including clinician notes and letters, referrals, health assessments/reports, available detention health documentation, Australian Immunisation Register (AIR) records, and/or school feedback/reports) using a standardised audit form created within the CAReHR™ database. This form included demographic information, details on detention (duration and location), reported trauma experience, symptoms, physical or mental

health diagnoses made during clinical review, and treatment and other supports provided. We included data from any time during or after held detention to capture the spectrum of impact.

We reported a range of clinical outcomes selected by clinical consensus, previous Australian refugee health research, and national guidelines for new arrival screening [20, 21]. We focused reporting on outcomes that were present at the time of first clinical assessment or became apparent in subsequent evaluation, documenting longer-term evolution of symptomatology where possible. Mental health diagnoses were included if they had been made by a child psychiatrist, paediatrician or psychologist, or where symptoms met Diagnostic and Statistical Manual of Mental Disorders (DSM-5) criteria [22]. Developmental disorders were included if diagnosed by a paediatrician or through a standardised multidisciplinary assessment (e.g. Autism Diagnostic Observation Schedule). Whilst standardised measurements for childhood trauma are often not validated for refugee/asylum seeker populations, our survey broadly covers domains of the Adverse Childhood Experience Rating Scores [23]. Absence of documentation was considered to be negative for symptoms and diagnosis. Six clinicians (ST, HG, GP, RH, TV, ILR) completed the initial data extraction; three investigators (ST, HG, GP) performed a secondary review to check for accuracy.

## Statistical methods

We extracted de-identified data from CAReHR to Microsoft Excel (Microsoft Corporation, Redmond, WA, USA) and used Stata™ (version 17.0) for statistical analysis. Categorical variables were summarised using number and percentage; normally distributed continous variables were summarised using mean and standard deviation while non-parametric variables were summarised using median and interquartile range. Results are shown for the overall cohort, and by two subgroups to examine differences based on location of detention (Nauru/Manus Island or Australia/Australian Territories). We used the Kruskal-Wallis test to compare duration of detention, and tests for equality of proportions to compare medical and mental health outcomes between the two subgroups.

## Results

In total, we identified 277 children (48% female) who had directly or indirectly (via parents) experienced held detention in Australia/Australian territories (n = 198) or Nauru or Manus Island (n = 79). Of these, 239 children (48% female) experienced held detention directly, including 46 in Nauru and one child on Manus Island. Thirty-one infants were born whilst in held detention (five on Nauru, 26 in Australian detention centres) and 38 children were born into families after release from detention (32/38 born to parents/families detained on Nauru/Manus Island). The entire cohort included children from 16 countries, speaking 16 languages; almost half (42%) were born in Iran (Table 1). Of children who had directly experienced held detention, two thirds (66%, n = 159/239) entered detention in 2013 (Fig 1).

## Pre-detention experience

Of 213 children born overseas, 132 (62%) reported experiencing a major traumatic event prior to arrival or during their journey to Australia (e.g. physical assault, kidnapping, boat capsized, war-related physical trauma). Eighteen children (8%) had experienced the death of an immediate family member due to conflict in their country of origin or during their migration journey.

**Table 1. Demographic details.**

| | Total (n = 277), n (%) | Australian/Australian Territories (n = 198), n (%) | Nauru/Manus Island (n = 79), n (%) |
|---|---|---|---|
| **Age* median (IQR), years** | 4.2 (0.7–7.8) | 4.8 (1.6–8.2) | 1.9 (0–7.0) |
| **Female** | 132 (48%) | 95 (48%) | 37 (47%) |
| **Country of Birth** | | | |
| **Australia** | 64 (23%) | 32 (16%) | 32 (41%) |
| Community Detention | 18 (28%) | 4 (13%) | 14 (44%) |
| Bridging Visa | 20 (31%) | 17 (53%) | 3 (9%) |
| Australian Detention Centre | 26 (41%) | 11 (34%) | 15 (47%) |
| **Nauru** (Australian Detention Centre) | 5 (2%) | 0 | 5 (6%) |
| **Overseas** | | | |
| Afghanistan | 7 (3%) | 7 (4%) | 0 |
| Bangladesh | 1 (< 1%) | 1 (1%) | 0 |
| Cyprus | 4 (1%) | 4 (2%) | 0 |
| India | 12 (4%) | 8 (4%) | 4 (5%) |
| Iran | 116 (42%) | 94 (47%) | 22 (28%) |
| Iraq | 12 (4%) | 10 (5%) | 2 (3%) |
| Lebanon | 5 (2%) | 4 (2%) | 1 (1%) |
| Malaysia | 20 (7%) | 15 (8%) | 5 (6%) |
| Myanmar | 4 (1%) | 4 (2%) | 0 |
| Nepal | 2 (1%) | 0 | 2 (3%) |
| Pakistan | 2 (1%) | 2 (1%) | 0 |
| Sri Lanka | 17 (6%) | 13 (7%) | 4 (5%) |
| Syria | 2 (1%) | 0 | 2 (3%) |
| Thailand | 2 (1%) | 2 (1%) | 0 |
| Yemen | 2 (1%) | 2 (1%) | 0 |
| **Preferred Language** | | | |
| Albanian | 1 (<1%) | 1 (1%) | 0 |
| Arabic | 47 (17%) | 36 (18%) | 11 (14%) |
| Burmese | 8 (3%) | 8 (4%) | 0 |
| Dari | 2 (1%) | 2 (1%) | 0 |
| Farsi | 140 (51%) | 95 (48%) | 45 (57%) |
| Hazaragi | 6 (2%) | 6 (3%) | 0 |
| Hindi | 1 (<1%) | 1 (1%) | 0 |
| Kurdish | 1 (<1%) | 1 (1%) | 0 |
| Kurdish–Faali | 2 (1%) | 2 (1%) | 0 |
| Malay | 5 (2%) | 0 | 5 (6%) |
| Nepali | 2 (1%) | 0 | 2 (3%) |
| Pashto | 2 (1%) | 2 (1%) | 0 |
| Rohingya | 19 (7%) | 15 (8%) | 4 (5%) |
| Sinhalese | 5 (2%) | 5 (3%) | 0 |
| Somali | 5 (2%) | 4 (2%) | 1 (1%) |
| Tamil | 31 (11%) | 20 (10%) | 11 (14%) |

* Age of 225 children at time of first entry into locked detention, age not available for 14 children.

## Held detention experience

Among 239 children who had direct experience of held detention, the median age at time of entry was 5.1 years (IQR 1.8 to 8.3 years). Children were typically detained in multiple

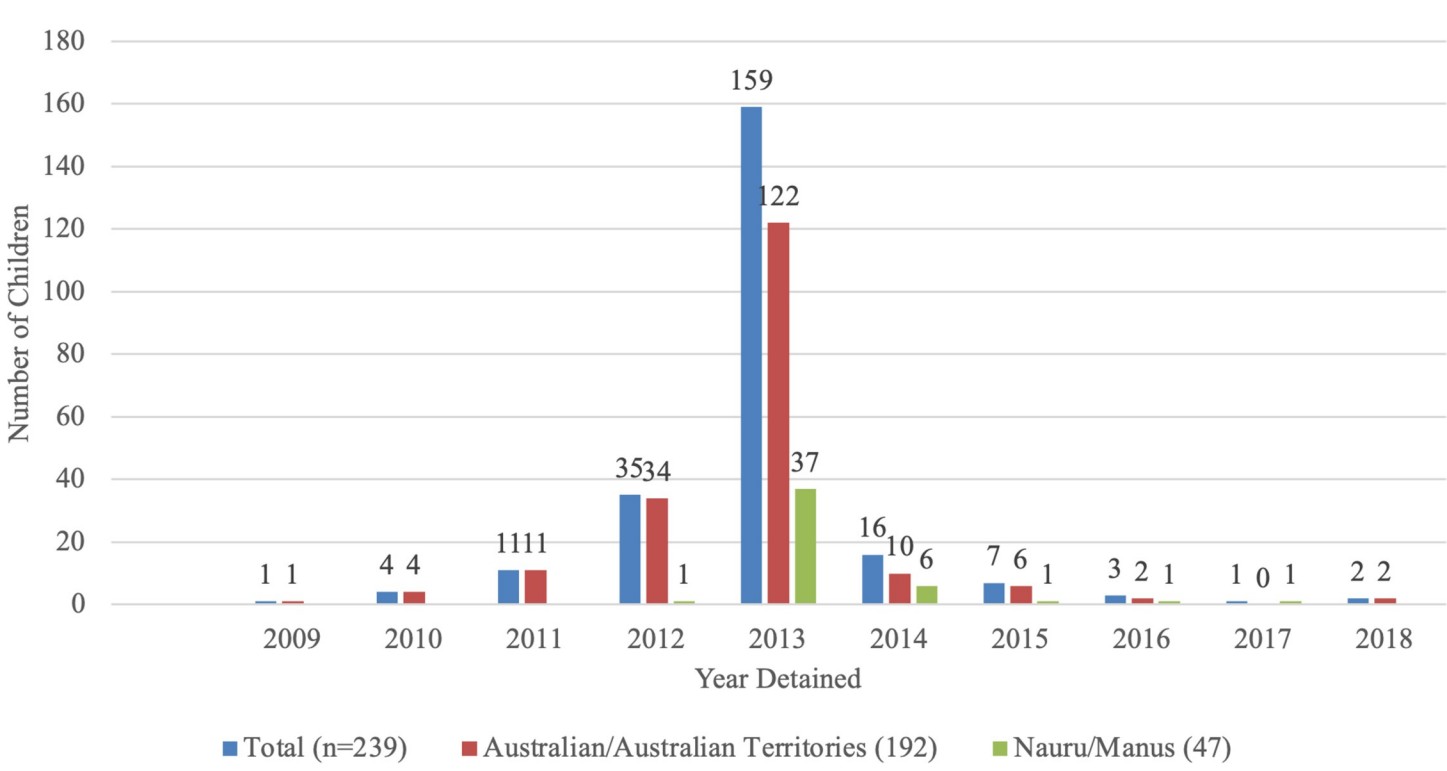

**Fig 1. Year and location of detention.**

detention centres (median 3 centres, range 1 to 8) for a median duration of 12 (IQR 5–19, range 1–67) months. Children who had been sent to Nauru/Manus Island were detained for a longer time than those who remained in Australian territories—median duration 51 (IQR 29–60) months compared to 7 (IQR 4–16) months ($p < 0.001$). At the time of their release from held detention, the median proportion of these children's lives spent in detention was 21% (IQR 7–42%). Median duration of CD (to end December 2021) for the cohort sent to Nauru/Manus Island was 42 (IQR 37–60) months with a combined duration of held and community detention for this cohort of 78 (IQR 59–99) months.

While in held detention, 21% children had been separated from at least one of their parents, in many cases for weeks or months, and at least 3 children were left alone in IDC while their parents were hospitalised. One quarter of children disclosed witnessing others self-harm, and 33% reported witnessing physical violence in IDC. Nineteen (8%) children who had been detained were referred for forensic paediatric medical evaluation following sexual assault concerns in detention.

## Health screening and immunisation

Australian refugee health guidelines recommend a comprehensive health assessment for all overseas-born asylum seeker children, and health screening was included in immigration detention contracts and services [20]. At time of first review at RCH, only 3 (1%) of 213 overseas-born children had completed recommended health screening. Overall, 8% (17/213) children had any results available with referral documentation (from either IHMS or community providers). After reviewing all available records and the AIR for 277 children, 29% were up to date with immunisation at time of first review at RCH. No record of vaccination had been entered into AIR during the period of detention for any children.

## Medical and developmental concerns

Close to two-thirds (60%, n = 167/277) of children were found to have a nutritional deficiency (Table 2). Low vitamin D was the most common deficiency, and was seen in children medically transferred from Nauru (50km from the equator). One in five (21%) of children required treatment for latent tuberculosis infection (LTBI) as per Australian guidelines after positive tuberculosis screening results, and none of these children had evidence of prior LTBI treatment in detention. A broad range of developmental and behavioural issues were identified, with some form of developmental concern identified for 75% children. Intellectual disability

**Table 2. Medical issues for children experiencing detention.**

| Medical Issue/Diagnosis | Total (n = 277) | | Australia/ Territories (n = 198) | | Nauru/Manus Island* (n = 79) | | Absolute Difference (%) | 95% CI**** | p-value**** |
|---|---|---|---|---|---|---|---|---|---|
| | **n** | **%** | **n** | **%** | **n** | **%** | | | |
| **Nutritional deficiency** | **167** | **60** | **120** | **61** | **47** | **59** | **2** | **-10 to 15** | **0.759** |
| Low vitamin D | 142 | 51 | 114 | 58 | 28 | 36 | 22 | 9 to 34 | 0.001 |
| Low B12 | 5 | 2 | 4 | 2 | 1 | 1 | 1 | -5 to 4 | 0.563 |
| Iron defiency without anaemia ** | 60 | 22 | 52 | 26 | 8 | 10 | 16 | 6 to 24 | 0.004 |
| Iron deficiency anaemia | 3 | 1 | 2 | 1 | 1 | 1 | 0 | -3 to 5 | 1 |
| Low zinc | 4 | 1 | 3 | 2 | 1 | 1 | 1 | -5 to 4 | 0.563 |
| **Infectious Disease** | **78** | **28** | **70** | **35** | **8** | **10** | **25** | **14 to 33** | **< .001** |
| Latent tuberculosis | 59 | 21 | 53 | 27 | 6 | 8 | 19 | 9 to 27 | < .001 |
| Strongyloidiasis | 7 | 3 | 5 | 3 | 2 | 3 | 0 | -4 to 7 | 1 |
| Stool parasites | 3 | 1 | 3 | 2 | | | n/a | | |
| Schistosomiasis | 2 | 1 | 2 | 1 | | | n/a | | |
| Scabies | 3 | 1 | 3 | 2 | | | n/a | | |
| Helicobacter pylori stool antigen positive | 9 | 3 | 7 | 4 | 2 | 3 | 1 | -6 to 5 | 0.691 |
| Fungal rash | 1 | | 1 | 1 | | | | | |
| **Development** | **207** | **75** | **155** | **78** | **52** | **66** | **12** | **1 to 24** | **0.039** |
| Parent concern regarding development | 105 | 38 | 72 | 36 | 33 | 42 | 6 | -6 to 19 | 0.353 |
| Learning difficulty*** | 83 | 30 | 65 | 33 | 18 | 23 | 10 | -2 to 20 | 0.102 |
| Vision concerns | 55 | 20 | 43 | 22 | 12 | 15 | 7 | -4 to 16 | 0.189 |
| Hearing concerns | 30 | 11 | 22 | 11 | 8 | 10 | 1 | -8 to 8 | 0.808 |
| Physical disability | 32 | 12 | 27 | 14 | 5 | 6 | 8 | -1 to 15 | 0.062 |
| Intellectual disability | 26 | 9 | 23 | 12 | 3 | 4 | 8 | 0 to 14 | 0.043 |
| Autism | 27 | 10 | 22 | 11 | 5 | 6 | 5 | -3.34 to 11.18 | 0.202 |
| **Other Medical Problems** | **266** | **96** | **193** | **97** | **73** | **92** | **5** | **-0.42 to 13.27** | **0.068** |
| Feeding difficulties | 111 | 40 | 71 | 36 | 40 | 51 | 15 | 2.18 to 27.46 | 0.022 |
| Secondary nocturnal enuresis | 54 | 19 | 42 | 21 | 12 | 15 | 6 | -4.76 to 14.79 | 0.254 |
| Constipation | 57 | 21 | 36 | 18 | 21 | 27 | 9 | -1.42 to 20.68 | 0.095 |
| Headaches | 27 | 10 | 18 | 9 | 9 | 11 | 2 | -5.06 to 11.36 | 0.610 |
| Encopresis | 15 | 5 | 13 | 7 | 2 | 3 | 4 | -3.03 to 8.90 | 0.201 |
| Epilepsy | 11 | 4 | 10 | 5 | 1 | 1 | 4 | -1.85 to 8.07 | 0.119 |
| Dental concerns | 118 | 43 | 85 | 43 | 33 | 42 | 1 | -11.88 to 13.40 | 0.876 |

*Child detained or from family detained on Nauru or Manus Island.

**Thalassemia trait/carrier excluded.

***e.g. language delay, concentration difficulty.

**** based on the test of equality of proportions.

and autism sprectrum disorder were prevalent, affecting 9% and 10% of children respectively; 13 of these children had both diagnoses. Feeding difficulties (e.g. inadequate intake, fussiness, food refusal) were commonly reported, particularly in infants from families detained on Nauru/Manus Island (51%, n = 40/79) and in infants born in held detention (52%, n = 16/31) (results not shown in tables).

## Mental health issues

Most children (62%, n = 171/277) had a psychiatric diagnosis and 52%, had psychiatric symptoms of concern (Table 3). The overall prevalence of clinician-diagnosed post-traumatic stress disorder (PTSD), anxiety, and depression was high, especially in the Nauru/Manus Island cohort. Almost half (43%) of the cohort had sleep difficulties, 27% reported nightmares and 10% of children had engaged in self-harm. Children in families detained on Nauru/Manus Island had a very high prevalence of mental health concerns across all domains (Table 3). Attachment issues with severe functional impact were diagnosed in 53% (n = 16/31) of infants born in detention, and 15% (n = 6/38) infants born in the community.

Parental mental health issues were prominent and impacted 54% (n = 150/277) of children, including 86% (n = 68/79) of those in the Nauru/Manus Island cohort. Twenty children (7%, n = 20/277) had parents who required admission to an inpatient psychiatric unit, of which most (80%, n = 16/20) were from the Nauru/Manus Island cohort. During or following held detention, 12% (n = 33/277) children experienced the breakdown of their parents' relationships, resulting in parent separation.

## Education, child protection and access to services

Interrupted education was common, with 46% (n = 92/199) school- and kindergarten-aged children having documented interruption to their education in detention, notably on

**Table 3. Mental health issues for children experiencing detention.**

| | Total (n = 277) | | Australian/ Australian Territory (n = 197) | | Nauru/Manus Island (n = 79) | | Absolute Difference (%) | 95% CI** | P-Value** |
|---|---|---|---|---|---|---|---|---|---|
| | N | % | N | % | N | % | | | |
| **Mental Health Issue** | **171** | **62** | **106** | **54** | **65** | **82** | **28** | **16 to 38** | **< .001** |
| PTSD* | 83 | 30 | 44 | 22 | 39 | 49 | 27 | 15 to 39 | < .001 |
| Anxiety | 121 | 44 | 71 | 36 | 50 | 63 | 27 | 14 to 39 | < .001 |
| Depression | 90 | 32 | 47 | 24 | 43 | 54 | 30 | 17 to 42 | < .001 |
| Behavior disorder | 112 | 40 | 73 | 37 | 39 | 49 | 12 | -1 to 25 | 0.067 |
| Attachment disorders | 68 | 25 | 30 | 15 | 38 | 48 | 33 | 21 to 45 | < .001 |
| **Psychiatric Symptoms** | **144** | **52** | **91** | **46** | **53** | **67** | **21** | **8 to 33** | **0.002** |
| Nightmares | 76 | 27 | 44 | 22 | 32 | 41 | 19 | 7 to 31 | 0.001 |
| Sleep difficulties | 120 | 43 | 72 | 36 | 48 | 61 | 25 | 12 to 37 | < .001 |
| School refusal | 35 | 13 | 11 | 6 | 24 | 30 | 24 | 14 to 35 | < .001 |
| Somatic complaints | 49 | 18 | 30 | 15 | 19 | 24 | 9 | -1 to 20 | 0.076 |
| Self-harm | 28 | 10 | 7 | 4 | 21 | 27 | 23 | 14 to 34 | < .001 |
| **Parent** | | | | | | | | | |
| Parent mental health issues | 150 | 54 | 82 | 41 | 68 | 86 | 45 | 33 to 54 | < .001 |
| Parent psychiatric admission | 20 | 7 | 4 | 2 | 16 | 21 | 19 | 11 to 29 | < .001 |

* PTSD–Post traumatic stress disorder.

** based on the test of equality of proportions.

Christmas Island, where schooling was often unavailable for the children in IDC, and in the Melbourne IDC, where children were not enrolled for months. No school-aged children in the Nauru/Manus Island cohort had had continuous schooling (results not shown in tables).

Child protection referrals/involvement were more common in children who had experienced detention in Nauru/Manus Island compared to those in Australian mainland/territories IDC (50%, n = 26/52 vs 19%, n = 52/277 children; difference 31%, CI 17–45, P < 0.001). However, formal investigations by the state child protection agency were hampered by jurisdictional issues (IDC are located on Commonwealth land and child protection legislation and services are state-based) and often did not progress beyond the intake process despite significant protective concerns.

At time of release from held detention, RCH clinicians were notified of the timing and location of community placement for only 5% (n = 11/239) children, and no health discharge assessments were received from IHMS before the first review following release. Most families contacted the RCH Immigrant Health Service directly to provide their updated contact details and continue medical care, or were reconnected through contact with other families from held detention.

Children in held detention were all registered with IHMS primary care, and 23% (n = 55/239) detained children in our cohort were recorded as having seen IHMS mental health services. The other main specialist service providers for these children during detention or following release were Foundation House (a trauma counselling service for refugees and asylum seekers)—in place for 42% (n = 115/277), and Child and Adolescent Mental Health Services—in place for 21%. Only 9% children reported receiving support from the Victorian Refugee Health Program (a program to support health and care coordination) after release from detention (results not shown in tables).

## Discussion

In this retrospective cohort of children who experienced prolonged held detention, we found a high prevalence of pre-arrival trauma, medical, mental health and developmental concerns. Interrupted education, child protection concerns and impacts from parental mental health issues and breakdown of family structure were common. Child and parent mental health issues, parent psychiatric admission and child protection concerns were all more frequent in children/families sent to RPCs. Most children in the Nauru/Manus Island cohort had mental health issues, and most also had parents with mental illness, reflecting the prolonged duration of detention and amplification of risks in the RPC conditions.

While Australian detention facilities provide medical services, our findings reveal a failure of basic preventive and public health care in IDC/RPC. Screening had not been completed, and children were identified with nutritional deficiencies, undiagnosed and untreated medical problems, including LTBI, and incomplete (and inadequately documented) routine childhood immunisations. Both onshore and offshore detention health records were difficult, and at times impossible, to obtain, and health discharge summaries were not available when children were released from detention. Our study supports the need for comprehensive health screening for asylum seekers, and documents a lack of screening and vaccination in Australian IDC/RPC. These findings are likely to represent the experiences of all children held in Australian detention.

Evidence from other case series and small cross-sectional studies have also shown high rates of psychiatric concerns in child asylum seekers who experienced Australian immigration detention, and significantly worse physical and mental health outcomes in comparison to child asylum seekers who were not detained [6–11, 24]. Our data support findings from other

studies examining the impact of Australian immigration detention. Paediatric asylum seekers in Western Australia were found to have complex trauma backgrounds, disrupted family units and negative health or education sequelae, compounded by detention experience [25]. At least 30 children from Nauru were diagnosed with 'pervasive refusal syndrome', with long term health implications [26], and The Australian Human Rights Commission 'National Inquiry into Children in Immigration Detention' (2014) documented adverse effects for 85% of children in detention [8]. Uncertainty arising from insecure visa status has also been shown to adversely affect health, Dari and Farsi speaking asylum seeker children in Australia have much higher rates of psychosocial problems compared to those with full refugee protection [27].

Reports from international settings reflect similar themes and the pervasive impact of held detention for children. Narratives from children experiencing immigration detention in Canada reflect the traumatic nature of immigration detention [28], and a survey of mental health in children held at a United States IDC found high levels of mental distress [29]. A systematic review on the impact of immigration detention on mental health identified a broad range of psychological disturbances in children, although clinical data were limited (3 studies, total 51 children/young people) [30]. Interviews of mothers of children in detention centres in the United States reflect high levels of mental distress [24] and high rates of mental health concerns in children separated from their parents [31].

Few studies have used in-depth audits [7, 25] with most previous studies investigating the physical and mental health of asylum seeker and refugee children using self-report, case reports, interviews and cross sectional survey methodology [32, 33]. A strength of this study is auditing clinical records across long-term clinical engagement with children and their families within a trusted service.

Witnessing violence, separation from parents and parental mental illness constitute major traumatising events for children, and emotional abuse and household dysfunction in childhood are identified risk factors for poor general and mental health in later life [34]. Children who experience detention are considered a high-risk population, requiring intensive, dedicated, preventive interventions when eventual settlement occurs [35]. Our study expands these findings, revealing a severity of impact that raises concern for long-term developmental, medical and mental health consequences in children who have experienced Australian detention [21, 36], and highlights the inappropriate conditions of immigration detention for children [37]. Other groups have also raised concern regarding the impact of Australian detention on children [38], and mandatory detention has been described as a form of child abuse by Australian pediatricians [39]. Peak bodies, including the Royal Australasian College of Physicians [40] and Australian Medical Association [41] have expressed alarm for the well-being of detained children, as have international organisations and researchers [36, 37, 42]. Arguments against detention extend beyond humanitarian and ethical arguments, to the financial cost of detention and long-term health cost for those affected [38].

Our study has limitations. The retrospective nature of an audit relies on both the information volunteered and documentation at the time of medical review. Often during clinical review there was a focus on acute health needs, the history was challenging and subject to difficulties with recall, exacerbated by poor parental mental health; and basic systems issues. Clinic visits from the Melbourne IDC to the Immigrant Health Service required advance notification and confirmation of appointments, transport bookings, security approvals and security clearance (pat downs, bag checks upon exit and re-entry). Despite a captive population, appointments were often missed (or children attended late) due to logistical and coordination difficulties, reducing time available for clinical care. Multiple security guards would accompany families and maintain line of sight at all times. Parents were often incapacitated by their own mental illness, contributing to challenges in assessing and providing healthcare for these

children. Most visits were completed with interpreting assistance, and there was secondary psychological impact for RCH interpreters, administrative and clinical staff, and at times, for transport and security staff. High turnover of immigration case managers, individual detention healthcare providers, immigration security staff, child protection staff, and subsequently, community-based case managers, also affected care delivery through loss of continuity, multi-layered bureaucracy and consequent communication issues. These factors complicated provision of care to a cohort in critical need of reliable healthcare [33]. However, these circumstances also mean that our reported findings are likely to be an underestimate. Longer term impact may not have been captured, and patients were lost to follow-up, especially in the initial period, when IDC transfers were frequent. Referral bias is likely, as our service sees refugee-background children with complex paediatric issues, however the commonality of issues over the years has been striking.

Ultimately, prioritisation of clinical care in a complex and changing policy environment meant that our study was not structured as prospective research. Our clinical concerns were documented and reported at the time, to detention health providers, to the Department of Home Affairs, and where relevant, to State Child Protection Services. However, it is also worth noting that when we started working with this cohort in 2012, we had no concept that detention could last for years, that children would not be enrolled in school in Australia, that families would be separated with so little regard for child wellbeing, and that child protection concerns would be so prominent and so difficult to address. We had not anticipated that we would be summarising our findings 10 years later, or that we would still be providing care for these same children with severe, long-term, developmental and mental health issues. After a decade, many of these children and families are still waiting for a decision on their primary protection claim, or in the case of those sent to Nauru/Manus Island, a defined migration pathway.

## Conclusion

This study provides clinical evidence for the adverse impact of held detention on children's physical and mental health, development and wellbeing. Health screening, immunisation and education access in held detention were inadequate. The environment of immigration detention was unsafe and harmful for children and must not be repeated. Policymakers must recognise the consequences of detention, and avoid detaining children and families, and Australian policies should not be emulated.

## Acknowledgments

We thank the RCH Immigrant Health Clinic team for their involvement in the study. We acknoweldge Tiba Maloof, Toni Mansfield, Tram Nguyen, Helen Milton and Alice Morgan for their assessments and support of families included in this cohort. We thank Dr. Monsurul Hoq for statistical support.

## Author Contributions

**Conceptualization:** Shidan Tosif, Hamish Graham, Andrea Smith, Tom Connell, Georgia Paxton.

**Data curation:** Shidan Tosif, Hamish Graham, Karen Kiang, Ingrid Laemmle-Ruff, Georgia Paxton.

**Formal analysis:** Shidan Tosif, Georgia Paxton.

**Investigation:** Shidan Tosif, Georgia Paxton.

**Methodology:** Shidan Tosif, Georgia Paxton.

**Project administration:** Shidan Tosif.

**Supervision:** Shidan Tosif, Tom Connell, Georgia Paxton.

**Validation:** Shidan Tosif.

**Writing – original draft:** Shidan Tosif, Andrea Smith.

**Writing – review & editing:** Shidan Tosif, Hamish Graham, Karen Kiang, Ingrid Laemmle-Ruff, Rachel Heenan, Thomas Volkman, Tom Connell, Georgia Paxton.

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
