## [Decision Letter · Decision Letter 0]

15 Nov 2022

PONE-D-22-29610Health of Children who experienced Australian Immigration DetentionPLOS ONE

Dear Dr. Tosif,

Thank you for submitting your manuscript to PLOS ONE. After careful consideration, we feel that it has merit but does not fully meet PLOS ONE’s publication criteria as it currently stands. Therefore, we invite you to submit a revised version of the manuscript that addresses the points raised during the review process. Please go though all the comments and suggestions made by both reviewers and  submit your revised manuscript by Dec 30 2022 11:59PM. If you will need more time than this to complete your revisions, please reply to this message or contact the journal office at plosone@plos.org. Please include the following items when submitting your revised manuscript:A rebuttal letter that responds to each point raised by the academic editor and reviewer(s). You should upload this letter as a separate file labeled 'Response to Reviewers'.A marked-up copy of your manuscript that highlights changes made to the original version. You should upload this as a separate file labeled 'Revised Manuscript with Track Changes'.An unmarked version of your revised paper without tracked changes. You should upload this as a separate file labeled 'Manuscript'.

We look forward to receiving your revised manuscript.

Kind regards,

Cesar Infante Xibille, Ph.D

Academic Editor

PLOS ONE

"No funding was used for this study."

4. Please amend the manuscript submission data (via Edit Submission) to include author Tom Connell.

Please do not edit]

Reviewers' comments:

Reviewer's Responses to Questions

**Comments to the Author**

1. Is the manuscript technically sound, and do the data support the conclusions?

Reviewer #1: Yes

Reviewer #2: Partly

2. Has the statistical analysis been performed appropriately and rigorously? 

Reviewer #1: Yes

Reviewer #2: Yes

3. Have the authors made all data underlying the findings in their manuscript fully available?

Reviewer #1: No

Reviewer #2: Yes

4. Is the manuscript presented in an intelligible fashion and written in standard English?

Reviewer #1: Yes

Reviewer #2: Yes

5. Review Comments to the Author

Reviewer #1: Summary of the revision and overall impression

The study tackles an important public health problem, the impact of held detention on the health and wellbeing of children and adolescents. This cross-sectional study used medical records from the Royal Children´s Hospital Immigrant Health Service, with the aim of describing the physical and mental health status of children who experienced held detention in Australia. One of the main strengths of the study is the data sources used (e.g., clinical records). For me, the strength of these type of data sources is that it can be seen as a more objective measure of health outcomes, when compared to self-reporting, which is a methodology often used in studies examining health outcomes of migrant populations. I would only suggest the authors to make a stronger case for the gap in the literature regarding the impact of detention centers on children’s health, by highlighting the methodology used. There are studies, mainly carried out in the United States, about the impact of detention centers on children´s health. However, as mentioned above, most use self-reported measures to evaluate the impact.

Major and minor issues

a) Major issues

- No major issues.

b) Minor issues:

1. General comments:

- I recommend language editing of the manuscript. The ideas presented in some paragraphs can be made clearer for the readers with minor language editing of the manuscript.

2. Introduction:

- It would be helpful for readers who are not familiarized with the Australian immigration system to have a better understanding in the introduction of the referral process to the Royal Children´s Hospital Immigrant Health Service. Why do children get referred to these health services? Are all children in detention receiving health services in this hospital or only those children who are referred by the immigration system? If this is the case, why some children get referred and why some children do not get referred to these health services? What are the criteria being used?

- It would also be helpful to know more about the differences between the Immigration Detention Centers on Australian mainland and the offshore Regional Processing Centers in Nauru and Manus Island, Papua New Guinea. What are the characteristics and processes that make these detention centers different from each other?

3. Methods:

- It would be useful for readers if authors described why they decided to analyze the health outcomes separately for the Immigration Detention Centers on Australian mainland and the offshore Regional Processing Centers in Nauru and Manus Island. This could be tied to the introduction suggestion of describing how are these immigration detention centers different.

4. Results:

- Include a statement like “results not shown in tables” for those results who are not included in the descriptive tables.

- I would suggest to only include the percentages in the parentheses, not both the percentages and the number of participants. Maybe just leave the n´s when the total n is different from what is previously stated in that paragraph. For example, delete here: “Almost half (43%, n=120/277) the cohort had sleep difficulties, 27% (n=76/277) reported nightmares and 10% (n=28/277) of children had engaged in self-harm”. But keep it here: “Attachment issues with severe functional impact were diagnosed in 53% (n = 16/31) of infants born in detention….”.

5. Discussion

- It is not clear in the manuscript at which point during the detention process children receive medical care in the Royal Children´s Hospital Immigrant Health Service. A section in the discussion about this would be helpful to readers to understand if the health outcomes are associated to their stay in the immigration detention centers or if they are a result of the migration process. For instance, if children are being evaluated in their first week after arrival to the detention centers, are the nutritional deficiencies associated to reduced access to nutritious food in the detention centers or are they associated to food insecurity in their countries of origin and during the migration process (e.g., before their arrival to the detention centers). I believe that integrating this into the discussion, and/or in the results section, could strengthen the argument the authors are making.

- Discuss the differences found when comparing health outcomes between the Immigration Detention Centers on Australian mainland and the offshore Regional Processing Centers in Nauru and Manus Island, Papua New Guinea. Why are some health outcomes more prevalent in Nauru vs. the Australian mainland detention centers, and vice versa? Are the conditions in these centers different? Could these conditions be impacting differently the health of children and families being detained?

- Discuss how generalizable are the study findings, based on the methodology implemented. Are these results generalizable only to participants in the study? Could the results be generalizable to other children held in detention in Australia or other parts of the world and why or why not?

6. Figures

- For Figure 1, I would suggest including axis captions for both the x and y axis, as well as a descriptive title.

Reviewer #2: A relevant issue about the health impact of detention on migrant children is addressed. Unfortunately, this situation is becoming more frequent in different parts of the world.

However, the following is recommended:

1) In the introduction more precisely the differences between the two places of detention and the reasons for these differences.

2) Include literature on the impact of detention on children's health in other countries, places, etc. To have a reference point of the results found in this investigation.

Method

3) Describe more the processes used for the selection of information.

4) Explain more clearly how mental health problems were detected, who was the specialist who made the diagnosis, with what criteria, etc.

5) The same comment for diagnoses of developmental disorders

6) Explain with greater precision the processes for the analysis of the information, not only the statistical analyzes used.

Discussion

7) The information in the following paragraphs should be considered as a hypothesis, since this information was not included in the results.

“Our experience working with children and families in detention also highlights challenges to service access and care. Visits from the Melbourne IDC to RCH required advance notification and confirmation of appointments, transport bookings, security approvals and security clearance (pat downs, bag checks upon exit and re-entry). Despite a captive population, appointments were often missed (or children attended late) due to logistical and coordination difficulties, reducing time available for clinical care. When children did attend RCH, multiple security guards would accompany families and maintain line of sight at all times, waiting outside the clinic doors during medical review.

Parents were often incapacitated by their own mental illness, contributing to challenges in

assessing and providing healthcare for these children. Most visits were completed with

interpreting assistance, and there was secondary psychological impact for RCH

interpreters, administrative and clinical staff, and at times, for transport and security staff.

High turnover of immigration case managers, individual detention healthcare providers,

immigration security staff, child protection staff, and subsequently, community-based

case managers, also affected care delivery through loss of continuity, multi-layered

bureaucracy and consequent communication issues"

6. PLOS authors have the option to publish the peer review history of their article (what does this mean?). If published, this will include your full peer review and any attached files.

Reviewer #1: No

Reviewer #2: No

---

## [Author Response · Author response to Decision Letter 0]

17 Jan 2023

RESPONSE TO REVIEWER’S COMMENTS

Reviewer #1: Summary of the revision and overall impression

The study tackles an important public health problem, the impact of held detention on the health and wellbeing of children and adolescents. This cross-sectional study used medical records from the Royal Children´s Hospital Immigrant Health Service, with the aim of describing the physical and mental health status of children who experienced held detention in Australia. One of the main strengths of the study is the data sources used (e.g., clinical records). For me, the strength of these type of data sources is that it can be seen as a more objective measure of health outcomes, when compared to self-reporting, which is a methodology often used in studies examining health outcomes of migrant populations. I would only suggest the authors to make a stronger case for the gap in the literature regarding the impact of detention centers on children’s health, by highlighting the methodology used. There are studies, mainly carried out in the United States, about the impact of detention centers on children´s health. However, as mentioned above, most use self-reported measures to evaluate the impact.

We thank the reviewer for this feedback and for highlighting the strengths of this manuscript and approach. We have added lines to the introduction (paragraph 5) and paragraph in the discussion (paragraph 6) highlighting this strength and the difference with other studies where self-reports and other methods are used.

Major and minor issues

a) Major issues

- No major issues.

b) Minor issues:

1. General comments:

- I recommend language editing of the manuscript. The ideas presented in some paragraphs can be made clearer for the readers with minor language editing of the manuscript.

We have made edits throughout the document to improve the readability of the manuscript.

2. Introduction:

- It would be helpful for readers who are not familiarized with the Australian immigration system to have a better understanding in the introduction of the referral process to the Royal Children´s Hospital Immigrant Health Service. Why do children get referred to these health services? Are all children in detention receiving health services in this hospital or only those children who are referred by the immigration system? If this is the case, why some children get referred and why some children do not get referred to these health services? What are the criteria being used?

We thank the reviewer for these comments and have added comments to explain the referral process (Methods paragraph 2). 

- It would also be helpful to know more about the differences between the Immigration Detention Centers on Australian mainland and the offshore Regional Processing Centers in Nauru and Manus Island, Papua New Guinea. What are the characteristics and processes that make these detention centers different from each other?

We have added comments in the introduction (paragraph 4,5) to describe the differences between these two types of detention centers, and their potential significance, in the introduction.

3. Methods:

- It would be useful for readers if authors described why they decided to analyze the health outcomes separately for the Immigration Detention Centers on Australian mainland and the offshore Regional Processing Centers in Nauru and Manus Island. This could be tied to the introduction suggestion of describing how are these immigration detention centers different.

We have added lines in the introduction to explain the rationale for analysing the different types of detention centers (paragraph 4,5) and methods (statistical methods, paragraph 1)

4. Results:

- Include a statement like “results not shown in tables” for those results who are not included in the descriptive tables.

- I would suggest to only include the percentages in the parentheses, not both the percentages and the number of participants. Maybe just leave the n´s when the total n is different from what is previously stated in that paragraph. For example, delete here: “Almost half (43%, n=120/277) the cohort had sleep difficulties, 27% (n=76/277) reported nightmares and 10% (n=28/277) of children had engaged in self-harm”. But keep it here: “Attachment issues with severe functional impact were diagnosed in 53% (n = 16/31) of infants born in detention….”.

Thank you for this suggestion – we have made changes accordingly throughout the results 

5. Discussion

- It is not clear in the manuscript at which point during the detention process children receive medical care in the Royal Children´s Hospital Immigrant Health Service. A section in the discussion about this would be helpful to readers to understand if the health outcomes are associated to their stay in the immigration detention centers or if they are a result of the migration process. For instance, if children are being evaluated in their first week after arrival to the detention centers, are the nutritional deficiencies associated to reduced access to nutritious food in the detention centers or are they associated to food insecurity in their countries of origin and during the migration process (e.g., before their arrival to the detention centers). I believe that integrating this into the discussion, and/or in the results section, could strengthen the argument the authors are making.

We have added lines to the methods, and discussion, to better reflect the timing of data collection and rationale.

- Discuss the differences found when comparing health outcomes between the Immigration Detention Centers on Australian mainland and the offshore Regional Processing Centers in Nauru and Manus Island, Papua New Guinea. Why are some health outcomes more prevalent in Nauru vs. the Australian mainland detention centers, and vice versa? Are the conditions in these centers different? Could these conditions be impacting differently the health of children and families being detained?

Thank you for highlighting this point. We have added points to the introduction and discussion to highlight the potential contributing factors to the differences. 

- Discuss how generalizable are the study findings, based on the methodology implemented. Are these results generalizable only to participants in the study? Could the results be generalizable to other children held in detention in Australia or other parts of the world and why or why not?

We have added comments to the demographics and discussion (paragraph 2), to discuss this point. 

6. Figures

- For Figure 1, I would suggest including axis captions for both the x and y axis, as well as a descriptive title.

This has been amended.

Reviewer #2: A relevant issue about the health impact of detention on migrant children is addressed. Unfortunately, this situation is becoming more frequent in different parts of the world.

We thank the reviewer for this feedback.

However, the following is recommended:

1) In the introduction more precisely the differences between the two places of detention and the reasons for these differences.

We have amended the introduction to better highlight differences and significance of the two places of detention.

2) Include literature on the impact of detention on children's health in other countries, places, etc. To have a reference point of the results found in this investigation.

We have added references to the introduction to other studies to provide greater context.

Method

3) Describe more the processes used for the selection of information.

4) Explain more clearly how mental health problems were detected, who was the specialist who made the diagnosis, with what criteria, etc.

5) The same comment for diagnoses of developmental disorders

6) Explain with greater precision the processes for the analysis of the information, not only the statistical analyzes used.

We thank the reviewer for these comments. We have made amendments to address points 3, 5 and 6. 

We feel point 4 is covered in “Variables” paragraph 2: Mental health diagnoses were included if they had been made by a child psychiatrist, paediatrician or psychologist, or where symptoms met Diagnostic and Statistical Manual of Mental Disorders (DSM-5) criteria.

Discussion

7) The information in the following paragraphs should be considered as a hypothesis, since this information was not included in the results.

“Our experience working with children and families in detention also highlights challenges to service access and care. Visits from the Melbourne IDC to RCH required advance notification and confirmation of appointments, transport bookings, security approvals and security clearance (pat downs, bag checks upon exit and re-entry). Despite a captive population, appointments were often missed (or children attended late) due to logistical and coordination difficulties, reducing time available for clinical care. When children did attend RCH, multiple security guards would accompany families and maintain line of sight at all times, waiting outside the clinic doors during medical review.

Parents were often incapacitated by their own mental illness, contributing to challenges in

assessing and providing healthcare for these children. Most visits were completed with

interpreting assistance, and there was secondary psychological impact for RCH

interpreters, administrative and clinical staff, and at times, for transport and security staff.

High turnover of immigration case managers, individual detention healthcare providers,

immigration security staff, child protection staff, and subsequently, community-based

case managers, also affected care delivery through loss of continuity, multi-layered

bureaucracy and consequent communication issues"

We thank the reviewer for this feedback. We have placed this paragraph in the limitations paragraph of the manuscript to provide context to this work and at the same time highlight some of the contextual limitations.

---

## [Decision Letter · Decision Letter 1]

23 Feb 2023

Health of Children who experienced Australian Immigration Detention

PONE-D-22-29610R1

Dear Dr. Shidan Tosif,

We’re pleased to inform you that your manuscript has been judged scientifically suitable for publication and will be formally accepted for publication once it meets all outstanding technical requirements.

Kind regards,

Cesar Infante Xibille, Ph.D

Academic Editor

PLOS ONE

Reviewers' comments:

Reviewer's Responses to Questions

**Comments to the Author**

1. If the authors have adequately addressed your comments raised in a previous round of review and you feel that this manuscript is now acceptable for publication, you may indicate that here to bypass the “Comments to the Author” section, enter your conflict of interest statement in the “Confidential to Editor” section, and submit your "Accept" recommendation.

Reviewer #1: All comments have been addressed

Reviewer #2: All comments have been addressed

2. Is the manuscript technically sound, and do the data support the conclusions?

Reviewer #1: Yes

Reviewer #2: Yes

3. Has the statistical analysis been performed appropriately and rigorously? 

Reviewer #1: Yes

Reviewer #2: Yes

4. Have the authors made all data underlying the findings in their manuscript fully available?

Reviewer #1: No

Reviewer #2: Yes

5. Is the manuscript presented in an intelligible fashion and written in standard English?

Reviewer #1: Yes

Reviewer #2: Yes

6. Review Comments to the Author

Reviewer #1: Authors have responded to previously sent comments and made the corresponding changes. I have no additional comments.

Reviewer #2: (No Response)

7. PLOS authors have the option to publish the peer review history of their article (what does this mean?). If published, this will include your full peer review and any attached files.

Reviewer #1: No

Reviewer #2: No

---

## [Editor Report · Acceptance letter]

27 Feb 2023

PONE-D-22-29610R1 

Health of children who experienced Australian immigration detention 

Dear Dr. Tosif:

I'm pleased to inform you that your manuscript has been deemed suitable for publication in PLOS ONE. Congratulations! Your manuscript is now with our production department. 

Kind regards, 

on behalf of

Dr. Cesar Infante Xibille 

Academic Editor

PLOS ONE